# Efficient Credit Assignment in Cooperative Multi-Agent Reinforcement Learning

## Abstract

Cooperative multi-agent reinforcement learning (MARL) algorithms are crucial in addressing real-world challenges wherein multiple agents collaborate to achieve common objectives. The effectiveness of these algorithms hinges on the accurate estimation of agent action values, typically attained through learning joint and individual action values. However, challenges arise due to the credit assignment problem since it is difficult to accurately attribute the global reward to the actions of individual agents, which limits sample efficiency. This paper introduces ECA, an episodic control-based method, to mitigate this limitation by directly evaluating and assigning individual agent credits. ECA leverages episodic memory to store and cluster past interaction experiences between agents and the environment. Building upon these experiences, we introduce an intrinsic reward signal, quantifying the individual agent credits to the joint goal. This proposed reward signal serves as a corrective measure to revise individual action values, thereby improving the accuracy of individual and joint value estimations. We evaluate our methodology on StarCraft multi-agent challenge (SMAC) and Google Research Football (GRF) tasks, demonstrating that our method significantly improves the sample efficiency of state-of-the-art cooperative MARL algorithms.

## 1 Introduction

Cooperative MARL has emerged as a pivotal paradigm for addressing complex real-world challenges such as autonomous driving (Shalev-Shwartz et al., 2016), traffic control(Wiering et al., 2000), robotics(Gupta et al., 2017; Wang et al., 2020a), health care(Abdellatif et al., 2021) and finace(Zhang et al., 2021a), where multiple agents need to achieve shared objectives by collaborating with teammates. Most MARL algorithms follow a dual-phase strategy including centralized training and decentralized execution (CTDE). The training of agents is centralized, typically using a centralized critic; while execution or decision-making occurs decentralized during deployment(Oliehoek et al., 2008; Kraemer & Banerjee, 2016; Rashid et al., 2020). Based on the CTDE paradigm, MARL algorithms(Sunehag et al., 2017; Rashid et al., 2020; Wang et al., 2021a) achieved great success in solving cooperative multi-agent tasks.

For CTDE paradigm, accurate credit assignment (Nguyen et al., 2018; Zhang et al., 2021a; Zhou et al., 2020) is a crucial problem for effective policy optimization. It allows agents to identify and reinforce actions that lead to desirable team success while discouraging actions that focus on individual success and harm the team proceeds. In contrast, inaccurate credit assignment introduces various issues that harm sample-efficient policy optimization, such as a lack of coordination among multiple agents (Sukhbaatar et al., 2016), sub-optimal joint policy(Zhou et al., 2023), and increased exploration challenges (Zheng et al., 2021). Therefore, it is important to explore the problem of accurate credit assignment.

However, ground truths for credit assignments are typically unavailable, and even domain experts find defining the credits both challenging and time-consuming. Several methods have been proposed to address these challenges in credit estimation. They can be divided into two categories, implicit credit assignment and explicit credit assignment. Implicit credit assignment focuses on learning associations between agents and observed rewards without explicitly linking rewards to individual actions (Lowe et al., 2017; Zhou et al., 2020; Wang et al., 2021b; Zhang et al., 2021b; Peng et al., 2021; Su et al., 2021). These methods typically

employ expressive value functions to coordinate agents. Unfortunately, existing studies (Rashid et al., 2020; Wang et al., 2021a; Chen et al., 2023) revealed that implicit credit assignment suffers from the limitation of monotonicity, which can lead to inaccurate estimations at the individual level. On the other side, explicit credit assignment directly computes and attributes credits to specific actions taken by each agent (Foerster et al., 2018; Wang et al., 2020b; Proper & Tumer, 2012). However, the accuracy of credit assignments remains limited due to the complexity of the environments, high-dimensional state spaces, and scarce background information.

To tackle the above limitations, in this paper, we introduce a simple yet effective episodic control method, named **ECA**, to conduct **E**fficient **C**redit **A**ssignment for more sample-efficient MARL policy optimization. Inspired by the paradigm of episodic control (Lengyel & Dayan, 2007; Pritzel et al., 2017; Blundell et al., 2016), the spirit of ECA is to utilize the *past interaction experiences* between agents and environments to help estimate the agent credits under certain states. However, different from existing episodic control works (Zheng et al., 2021; Na et al., 2024) that focus on measuring the global credits, ECA supports more fine-grained and agent-level explicit credit assignment. Overall, ECA contains two phases, episodic memory construction and episodic control. In the first phase, we reduce the dimension of concrete global states and discretize them into unique and countable abstract states, thus concrete states can be naturally grouped based on abstract topological similarity. After that, leveraging statistics of past trajectories with the same abstract states, we calculate the average expectation of episodic returns for each agent, serving as individual credits. Then we integrate these credits to correct individual values during policy optimization.

The essence of ECA lies in the direct measurement and correction of the individual agent's impact on the team, thereby improving the sample efficiency of MARL policy optimization. Experimental evaluations have been conducted on SMAC (Samvelyan et al., 2019) and GRF (Kurach et al., 2019) tasks involving comparisons with existing MARL algorithms and episodic control-based methods. The results convincingly demonstrate that ECA significantly improves sample efficiency. The ECA code repository is publicly available for further reproduction[1].

In summary, this work makes the following contributions:

- The paper introduces ECA, a new episodic control method designed to improve sample efficiency in cooperative MARL by enabling more accurate credit assignment to individual agents.

- ECA provides fine-grained, explicit credit assignments at the agent level, allowing for more precise policy optimization.

- The paper validates ECA through experiments on standard MARL benchmarks, namely SMAC and GRF. Results show that ECA outperforms existing MARL algorithms and episodic control-based methods, demonstrating significant improvements in sample efficiency.

- We make the ECA code repository publicly available, promoting transparency and enabling other researchers to reproduce and build upon their work.

## 2 Preliminaries

### 2.1 Multi-agent Reinforcement Learning

MARL builds upon the Decentralized Partially Observable Markov Decision Processes (Dec-POMDPs) (Oliehoek & Amato, 2016) framework where multiple agents interact within an environment characterized by partial observability. A Dec-POMDP is a tuple $\langle \mathcal{I}, \mathcal{S}, \mathcal{A}, \mathcal{P}, \mathcal{R}, \Omega, \mathcal{O}, n, \gamma \rangle$, where $\mathcal{I}$ is a set of $n$ agents $(I_0, ..., I_i, ..., I_{n-1})$ which take the observations $\mathbf{o}^t$ as inputs. $\mathcal{S}$ represents the state space constructed by states across different time steps $s^t$, where $t$ is the time step. $\mathcal{A}$ denotes the action space of the agents thus the joint action $\mathbf{a} \in \mathcal{A}^n$. $\mathcal{P}(s^{t+1}|s^t, \mathbf{a}^t)$ denotes the transition function of state $s^t$ and joint action $\mathbf{a}^t$, $\mathcal{R}(s^{t+1}|s^t, \mathbf{a}^t)$ is the reward function. $\mathbf{o}^t = \mathcal{O}(s^t, I)$ is the observations of all agents, and $\gamma$ is the discount factor. Dec-POMDPs encapsulate the inherent complexity of decentralized decision-making, where agents

---

[1] https://anonymous.4open.science/r/ECA-Repo

make informed choices based on their local observations while considering the actions and observations of their peers. And the interaction trajectory can be represented as:

$$(s^0, \mathbf{o}^0, \mathbf{a}^0, r^0), ..., (s^t, \mathbf{o}^t, \mathbf{a}^t, r^t), ..., (s^T, \mathbf{o}^T, \mathbf{a}^T, r^T) \tag{1}$$

Centralized Training and Decentralized Execution (CTDE) (Bernstein et al., 2002; Oliehoek et al., 2008; Gupta et al., 2017) is a prominent paradigm in MARL that addresses the challenges of coordinating multiple agents while optimizing their policies. The key characteristic of CTDE lies in its two-phase strategy, where training is performed centrally while execution or decision-making occurs in a decentralized manner during deployment. During the training process, at time-step $t$, the whole team cooperates to find the optimal joint action-value function $Q_{tot}^*(s, \mathbf{a})$:

$$Q_{tot}^*(s^t, \mathbf{a}^t) = r(s^t, \mathbf{a}^t) + \gamma \mathbb{E} \left[ \max Q_{tot}^*(s^{t+1}, \mathbf{a}^{t+1}) \right] \tag{2}$$

During execution, the learned policies are employed independently, allowing each agent to make decentralized decisions based on local observations. CTDE balances centralized knowledge aggregation and decentralized decision-making, providing a practical framework for addressing complex coordination challenges in MARL scenarios. However, the CTDE framework typically relies on global rewards obtained by the team for value estimation and policy optimization, which could not provide accurate agent-level credit assignments.

## 2.2 State Abstraction

The State Abstraction method incorporates a comprehensive episodic memory that facilitates a more advanced analysis of past experiences(Li et al., 2023). By discretizing the continuous state space into finite grids across each dimension, states located in the same grid will have the same abstract representation and can be clustered automatically based on abstract topological similarity. This approach also reduces the complexity of storage and retrieval.

Specifically, a high-dimensional state space that contains infinite concrete states, each concrete state $s$ can be represented as $(\phi^0, ..., \phi^d, ...)$. Each component $\phi^d$ is a scalar value and has lower and upper boundaries $\phi^d \in [l_d, u_d]$, which is determined by the environment. We split each dimension into $N$ equal intervals, and the concrete value $\phi^d$ falls into an interval $\eta_d$:

$$\eta_d = \lfloor \frac{u_d - l_d}{N} * (\phi^d - l_d) \rfloor \tag{3}$$

In this way, a high-dimensional concrete state $s$ can be represented as an abstract state $\hat{s} = (\eta_0, \ldots, \eta_d, \ldots)$, which encodes each dimension of the concrete state to an integer and is easy to count. Similar concrete states are grouped into the same abstract state if their concrete values fall within the same intervals. This approach allows for measuring the effects of different actions by agents within the same abstract state, enabling quantifying their contributions at the abstract state level.

Given a two-dimensional concrete state (0.32, 0.73), each feature ranges from 0 to 1. We set grid number N = 5, and then each dimension is split into five intervals: $\eta_1 = [0, 0.2)$, $\eta_2 = [0.2, 0.4)$, $\eta_3 = [0.4, 0.6)$, $\eta_4 = [0.6, 0.8)$, $\eta_5 = [0.8, 1.0]$, which can be represented as 1, 2, 3, 4, and 5, respectively. 0.32 falls in $\eta_2$, 0.73 falls in $\eta_4$. Thus the original state (0.32, 0.73) can be represented in an abstract state (2, 4). The abstract state is easier to count based on their occurrences than concrete states.

## 2.3 Reward Confidence Score

The reward confidence score efficiently measures the episodic returns resulting from the states. The computation of the reward confidence score is based on the statistic of episodic returns. Given the trajectory $\psi_k$:

$$\psi_k = \{(\hat{s}^0, \mathbf{o}^0, \mathbf{a}^0, r^0), ..., (\hat{s}^t, \mathbf{o}^t, \mathbf{a}^t, r^t), ..., (\hat{s}^T, \mathbf{o}^T, \mathbf{a}^T, r^T)\} \tag{4}$$

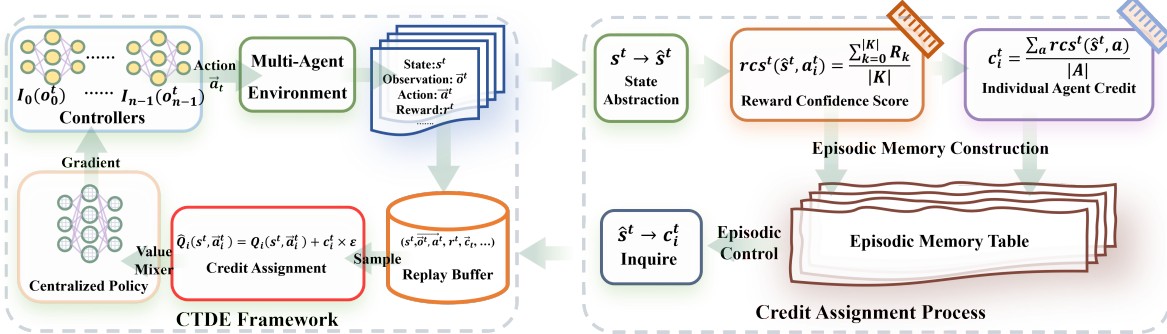

Figure 1: An overview of ECA workflow.

$T$ is the length of the trajectory $\psi_k$, note $\hat{s}^t$ is an abstract state at time step $t$, the episodic return $\mathcal{E}_k$ of the trajectory $\psi_k$ is computed as follows:

$$\mathcal{E}_k = \sum_{t=0}^{T} r^t \tag{5}$$

Thus the reward confidence score $rcs^t(\hat{s}^t)$ of global state $\hat{s}^t$ can be computed by the following method:

$$rcs^t(\hat{s}^t) = \frac{1}{|\Psi|} \sum_{k=0}^{|\Psi|} \mathcal{E}_k, \hat{s}^t \in \Psi \tag{6}$$

where $\Psi$ is the collection of trajectories that encountered the state $\hat{s}^t$. The reward confidence score $rcs^t(\hat{s}^t)$ represents a *quality* measurement of global state $\hat{s}^t$. Specifically, a higher $rcs^t(\hat{s}^t)$ indicates that the team earns more returns when encountering $\hat{s}^t$.

In previous reward confidence score-based policy optimization (Li et al., 2023), $rcs^t(\hat{s}^t)$ is used as an explicit signal and added to the global reward for enhancing the joint action value $Q_{tot}^*$:

$$
\begin{aligned}
Q_{tot}^*(s^t, \mathbf{a}^t) = & \, r(s^t, \mathbf{a}^t) \\
& + \gamma \mathbb{E} \left[ \max Q_{tot}^*(s^{t+1}, \mathbf{a}^{t+1}) \right] \\
& + rcs^t(\hat{s}^t)
\end{aligned}
\tag{7}
$$

## 3 Methodology

This section introduces the details of the ECA methodology. The core concept of ECA is to estimate agent contributions based on past experiences in various states. Specifically, we calculate the episodic returns achieved for actions taken by each agent in particular states. ECA comprises two main steps: episodic memory construction and episodic control. Figure 1 illustrates the ECA workflow, built upon the classic CTDE framework.

In the episodic memory construction phase, ECA begins by discretizing the concrete states into unique and countable abstract states (*i.e.*, State Abstraction, as introduced in Section 2.2). Specifically, the states are grouped and clustered by their abstract topological similarity. Following this state abstraction, agent credits for each abstract state are calculated (i.e., $C_i^t$) and stored in an episodic memory table. In the episodic control phase, these stored agent credits are retrieved to efficiently adjust individual action values and enhance centralized policy optimization (Schaul et al., 2015).

---

**Algorithm 1** Episodic Memory Construction

---

**Input:** $(I_0, ..., I_i, ..., I_{n-1})$: a set of $n$ agents, $\mathbf{o}^t$: observation of each agent at time step $t$, $E$: Dec-POMDP environment, $s$: global state of the environment, $\mathcal{M}$: episodic memory, *Steps*: number of time steps.
**Output:** $\mathcal{B}$: Replay Buffer with episodic data and our individual credits.

1: Initialize: $\mathcal{B}, \mathcal{F} \leftarrow \emptyset, t = 0$
2: **while** $t < Steps$ **do**
3:     $s^t, \mathbf{o}^t \leftarrow E$.Reset()
4:     $\hat{s} \leftarrow \mathcal{M}$.State_Abstraction($s^t$)                    ▷ Convert concrete states to unique labels.
5:     $\mathbf{a}^t \leftarrow \mathcal{I}$.Agents_Action_Selection($\mathbf{o}^t$)          ▷ Decentralized agents select actions for execution.
6:     $s^{t+1}, \mathbf{o}^{t+1}, r^t \leftarrow E$.Step($\mathbf{a}^t$)
7:     $\langle c_0, ..., c_i, ..., c_{n-1} \rangle \leftarrow \mathcal{M}$.Update($\hat{s}, \mathbf{a}^t, r^t$)          ▷ Calculate individual credit of all the agents.
8:     $\mathcal{B}$.append($s^t, \mathbf{o}^t, \mathbf{a}^t, r^t, \langle c_0, ..., c_i, ..., c_{n-1} \rangle$)          ▷ Store the individual credits into replay buffer $\mathcal{B}$.
9:     $t \leftarrow t + 1$
10: **end while**

---

## 3.1 Episodic Memory Construction

To measure agent credit, we quantify the contribution of each agent's actions in specific states based on past experiences. However, the concrete states are high-dimensional and continuous, leading to an infinite number of states, which poses challenges for state-based credit analysis. To mitigate this, we adopt a topological state clustering method for state abstraction, which discretizes the states and merges similar concrete states together. In this way, we divide infinite state space into finite groups and perform the credit analysis of agent actions on each group.

**State-based Credit Measurement.** The past experiences, represented by $K$ trajectories (*i.e.*, all the concrete states, actions, and rewards in a trajectory), provide valuable information for assessing the contribution of each agent's actions. In MARL, each trajectory can be represented as $\psi = \{(\hat{s}^0, \mathbf{o}^0, \mathbf{a}^0, r^0), ..., (\hat{s}^t, \mathbf{o}^t, \mathbf{a}^t, r^t), ...\}$, where $\hat{s}^t$ is an abstract state (based on interval abstraction) of concrete state $s^t$ shared by all the agents at time step $t$, $\mathbf{o}^t$ contains the respective observation of each agent, and $\mathbf{a}^t$ is the joint action of the team at time step $t$. For each trajectory $\psi$, the episodic return $\mathcal{E}_\psi$ is calculated by the sum of the rewards across all the time steps, details can be found in Section 2.3. For each action $a_i$ (taken by the $i^{th}$ agent) under a specific state $\hat{s}$, we measure its *contribution* by counting the average of the episodic return of the trajectories that cover the state $\hat{s}$. The contribution is represented by the reward confidence score:

$$rcs(\hat{s}, a_i) = Avg(\{\mathcal{E}_\psi | \psi \in \Psi \wedge cover((\hat{s}, a_i), \psi)\}) \tag{8}$$

where $\Psi$ represents all the trajectories, and *cover* indicates that the trajectory $\psi$ includes the state $\hat{s}$ and the action $a_i^t$ taken by the $i^{th}$ agent in that state. Intuitively, the contribution of the action $a_i^t$ under the state $\hat{s}$ is determined by the average episodic return of the trajectories that include this action in that state.

As an agent may take different actions in the same abstract state across various trajectories, ECA calculates the reward confidence scores for all actions within the state to estimate each agent's impact on state $\hat{s}$. We assess the individual credit of the $i^{th}$ agent under a specific state $\hat{s}$:

$$c_i(\hat{s}) = Avg(\{rcs(\hat{s}, a_i) | a_i \in \mathcal{A}\}) \tag{9}$$

where $\mathcal{A}$ represents all possible actions of the $i^{th}$ agent in state $\hat{s}$. With the individual credit $c_i$, we can measure the influence of the agents on the episodic returns. A higher individual credit reflects a stronger impact of the individual decision-making on the team.

Algorithm 1 summarizes the processes of Episodic Memory Construction. We utilize episodic data to measure individual credits first, then store them in a tabular episodic memory. Specifically, we transform each concrete state $s^t$ into an abstract state $\hat{s}$ via topological state abstraction (line 4). Following the action selection for each agent and decentralized execution (line 7), we input all abstract states, actions, and rewards into the episodic memory module to compute the individual credit of each agent at every time step (*i.e.*, under the state at the specific time step). Subsequently, individual credits are deposited into the replay buffer $\mathcal{B}$ for subsequent policy optimization (line 8).

---

**Algorithm 2** Episodic Control-Based Credit Assignment

---

**Input:** $\mathcal{I}$: decentralized agents, $n$: number of agents, $\mathcal{P}$: centralized policy, $\mathcal{B}$: replay buffer, $\mathcal{Z}$: batch size, $\gamma$: discount factor, $\epsilon$: scaling factor in ECA.
**Output:** $Loss$: Optimal Centralized Policy.
 1: Initialize: $Loss = 0$
 2: **while** $u < \mathcal{Z}$ **do**
 3: $\quad$ $s^u, \mathbf{o}^u, \mathbf{a}^u, r^u, \langle c_0^u, ..., c_i^u, ..., c_{n-1}^u \rangle \leftarrow \mathcal{B}.\text{Sample}()$
 4: $\quad$ **for** $i = 0 \rightarrow n - 1$ **do**
 5: $\quad\quad$ $Q_i^u(o_i^u, a_i^u) \leftarrow \mathcal{I}_i.\text{Individual\_Estimation}(o_i^u, a_i^u)$
 6: $\quad\quad$ $\hat{Q}_i^u(o_i^u, a_i^u) \leftarrow Q_i^u(o_i^u, a_i^u) + c_i \times \epsilon$ $\qquad\qquad\qquad$ ▷ Revision of individual values by credits.
 7: $\quad$ **end for**
 8: $\quad$ $\hat{Q}_{tot}(s^u, \mathbf{a}^u) \leftarrow \mathcal{P}.\text{Value\_Integration}(\hat{Q}_0^u, ..., \hat{Q}_{n-1}^u)$ $\qquad$ ▷ Value integration operation, *e.g.*, QMIX.
 9: $\quad$ $\delta^u \leftarrow r^u + \gamma \max_{\mathbf{a}'} Q_{tot}(s', \mathbf{a}') - \hat{Q}_{tot}(s^u, \mathbf{a}^u)$ $\qquad\qquad$ ▷ $s', \mathbf{a}'$ are the next state and action of $s^u, \mathbf{a}^u$.
10: $\quad$ $Loss = Loss + \frac{1}{2}(\delta^u)^2$
11: $\quad$ $u \leftarrow u + 1$
12: **end while**

---

## 3.2 The episodic memory structure of ECA

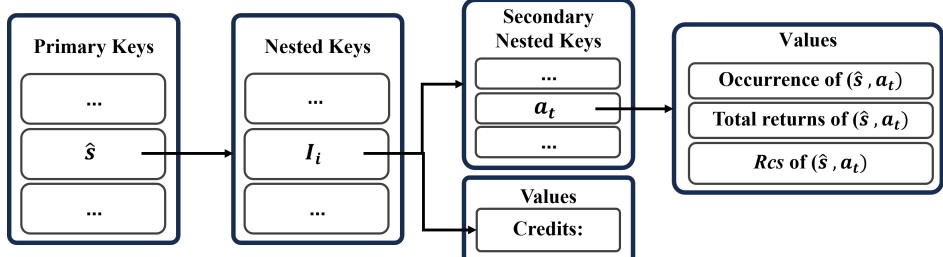

Figure 2: The specific structure of episodic memory in ECA. Our memory is a nested key-value table that records the agent credits and the information for credit computation.

Figure 2 illustrates the structure of episodic memory within ECA. Episodic memory in ECA is structured as a nested key-value table. Specifically, it utilizes an abstract state as the primary key, facilitating precise retrieval and efficient memory operations. Nested within are agent IDs, with each agent having a corresponding credit value. Furthermore, the secondary nested keys represent the actions undertaken by each agent. These actions are associated with three values: the occurrence of abstract state and agent action pairs $(\hat{s}, a_i^t)$, the cumulative episodic returns within the trajectory $\psi$ containing $(\hat{s}, a_i^t)$, and a reward confidence score denoting the average episodic returns, detailed in Equation 8.

During memory construction (as outlined in Algorithm 1), interaction experiences are gathered at each time step $t$. Initially, the concrete state $s_t$ is transformed into an abstracted state $\hat{s}$ (line 4). Subsequently, occurrences of each abstract state and agent action pair $(\hat{s}, a_i^t)$ are recorded, alongside updates to corresponding cumulative episodic returns and reward confidence scores. With this information stored in episodic memory, agent credits are recalculated based on their actions' aggregate reward confidence scores, following Equation 9. This process ensures accurate approximation and retrieval of agent credits, crucial for further policy optimization (as outlined in Algorithm 2), enabling efficient episodic control and credit assignment within ECA.

## 3.3 Episodic Control

After measuring credit based on past trajectories, we apply the estimated credit in episodic control. The individual credits are used as intrinsic rewards for each agent through reward shaping (Ng et al., 1999; Burda et al., 2019; Zheng et al., 2021; Na et al., 2024):

$$\hat{Q}_i(o_i^t, a_i^t) = Q_i(o_i^t, a_i^t) + c_i \times \epsilon \qquad\qquad (10)$$

Table 1: The hyperparameters and policy network architecture of ECA.

| Items | Values |
| --- | --- |
| Replay Buffer Size | 5000 |
| Batch Size | 32 |
| Discount Factor | 0.99 |
| Individual Agent Optimizer | Adam |
| Individual Agent Learning Rate | 5e-4 |
| Centralized Policy Optimizer | Adam |
| Centralized Policy Learning Rate | 5e-4 |
| Hidden Layer Size | 64 |

where $Q_i(o_i^t, a_i^t)$ represents the individual action value of the $i^{th}$ agent at time step $t$, and $\epsilon$ is a scalar to control the value scale of shaping reward. Note that the $o_i^t$ can either be the current or historical observations. Using current observations allows agents to react promptly to the environment's current state and the actions of other agents, while historical observations provide valuable context and enable agents to learn from past experiences. In ECA, individual agents estimate the action values using the respective historical observations. After the reshaping of individual credits, the new individual value $\hat{Q}_i$ will be fed into a value integration module, such as mixture network in QMIX (Rashid et al., 2020), and then the value of $Q_{tot}^*(s^{t+1}, \mathbf{a}^{t+1})$ and $Q_{tot}^*$ will be updated. Finally, we can obtain a revised joint action value for loss evaluation for centralized policy network update.

Algorithm 2 presents the details of Episodic Control. We enhance policy optimization by incorporating individual credits into original agent values (line 6). These adjusted individual agent values are then aggregated to compute the joint-action values (line 8). Consequently, the centralized policy is trained by minimizing discrepancy between estimated joint-action values and the temporal difference (TD) target, where the estimated joint-action values have been modified using individual credits.

## 4 Evaluation

### 4.1 Experiment setup

**Dataset.** We evaluate ECA on commonly used tasks SMAC (Samvelyan et al., 2019) and GRF (Kurach et al., 2019). SMAC and GRF provide various challenging multi-agent tasks, in which the agents need to experience millions of high-dimensional concrete states to learn a good policy. Specifically, we select six SMAC maps (Samvelyan et al., 2019) that vary by task difficulties, including an *Easy* task *1c3s5z*, two *Hard* tasks *3s_vs_5z* and *5m_vs_6m*, and three *Super Hard* tasks *3s5z_vs_3s6z*, *5s10z*, and *MMM2*, in where diverse agents (*e.g.*, *s* denotes *Stalkers* in the game) engage in team battles against common enemies. For the GRF task, we select academy_3_vs_1_with_keeper (*3_vs_1_WK*), academy_counterattack_easy (*CA_Easy*), and academy_counterattack_hard (*CA_Hard*), in which different numbers of players are controlled by the centralized policy to score in a football game under different conditions (Kurach et al., 2019).

**Baseline.** With the above tasks, we compare ECA with Curiosity-Driven Exploration (EMC) (Zheng et al., 2021) and Efficient Episodic Memory Utilization (EMU) (Na et al., 2024), representative episodic control-based methods designed to improve the sample efficiency of existing cooperative MARL algorithms. EMC stores states in episodic memory, incorporates a curiosity module to measure the novelty of states, and employs this assessment as a guide for exploration. Conversely, EMU focuses primarily on desired (highly-rewarded) state transitions, employing an episodic memory to store past experiences and measure corresponding expected returns as intrinsic rewards. We select Qplex (Wang et al., 2021a) as the base algorithm for EMC, EMU, and ECA to enable an adequate comparison. Moreover, the default value integration method is Qatten (Yang et al., 2020a). The program of EMC and EMU is from the official and open code repositories, and the implementation of ECA is based on the PyMARL benchmark(Samvelyan et al., 2019).

**Configurations.** ECA is highly supplementary and straightforward, requiring no additional adaption on various MARL algorithms, rendering promising applicability and potential for solving more multi-agent tasks. For a fair comparison, we follow previous works (Zheng et al., 2021; Na et al., 2024) to set hyperparameters and network architecture, as listed in Table 1. Besides, ECA performs dimension reduction of concrete global states by multiplying a Gaussian random projection matrix, which makes smaller concrete global states and efficient state abstraction. These processed state vectors have a $(1, 24)$ shape on all the tasks. Subsequently, we set $N$ in Equation 3 as 5. The scaling factor $\epsilon$ for shaping reward values in Equation 10 to 0.001 aligns with the values utilized in prior methods (Na et al., 2024). The other parameter choices also mirror the settings in previous works (Na et al., 2024), including the policy network architecture, learning rate, size of replay buffer, etc. We repeat each experiment five times to counteract randomness.

## 4.2 Evaluation of ECA on large-scale multi-agent tasks

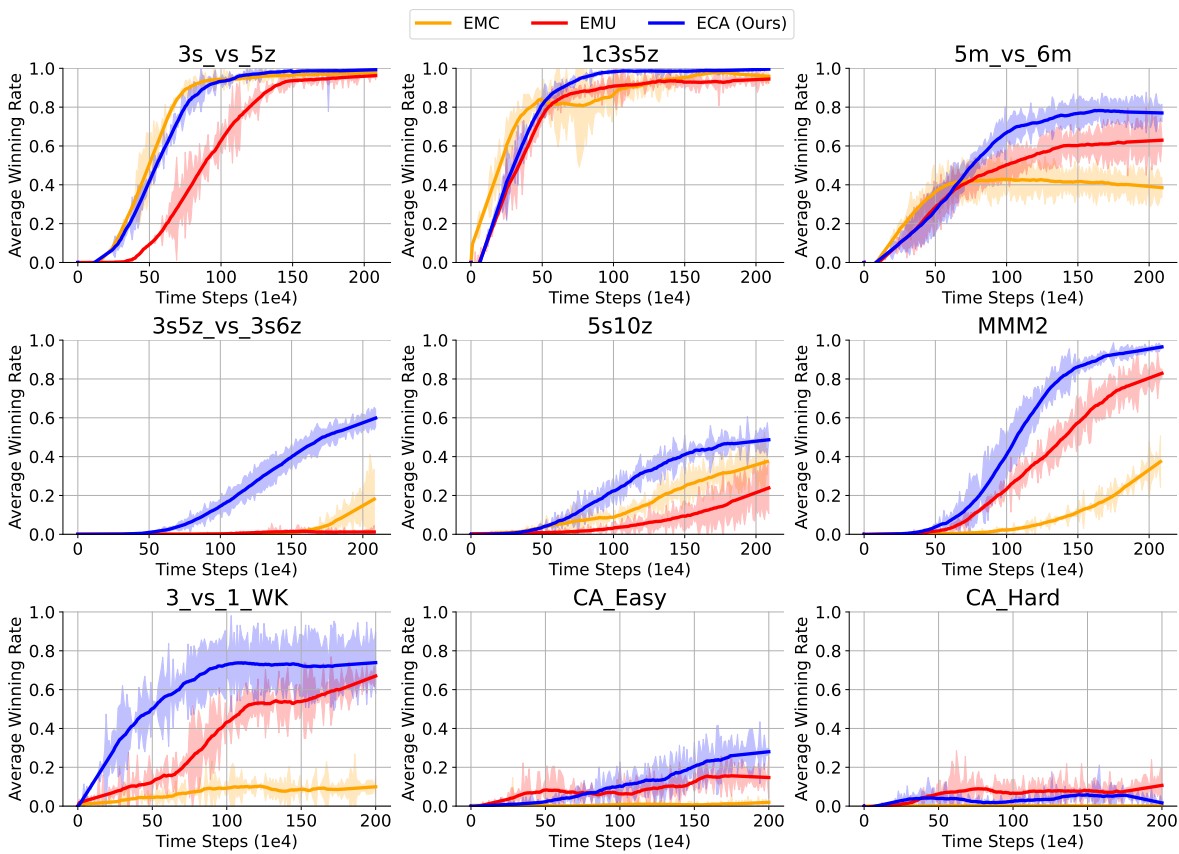

Figure 3: The evaluation of ECA and the state-of-the-art MARL episodic control methods.

Figure 3 illustrates the trends in team winning rates during policy optimization by EMC, EMU, and our ECA. Table 2 lists the average winning rates achieved by policies trained after 2 million time steps. In SMAC tasks, while the winning rates of the three methods on the *Easy* task *1c3s5z* and the *Hard* task *3s_vs_5z* are close, ECA significantly outperforms EMC and EMU on *5m_vs_6m*, *3s5z_vs_3s6z*, *5s10z*, and *MMM2*, which are more challenging due to the number of agents and the strength of the enemies. In GRF tasks, ECA achieves 0.736 and 0.276 winning rates and outperforms other methods on *3_vs_1_WK* and *CA_Easy*, respectively. All methods, including EMC, EMU, and ECA, perform worse on the *CA_Hard* task in GRF due to the task difficulties. In summary, ECA enables a more sample-efficient MARL policy optimization on most designated tasks compared to the baselines.

Table 2: The average winning rates and standard deviations by EMC, EMU, and our ECA. Note that bold marks denote the highest winning rate achieved in the corresponding tasks.

| Domains | Tasks | EMC | EMU | ECA (Ours) |
|---|---|---|---|---|
| SMAC | 1c3s5z | 0.961±0.011 | 0.943±0.011 | **0.994±0.001** |
| | 3s__vs__5z | 0.977±0.005 | 0.964±0.003 | **0.991±0.001** |
| | 5m__vs__6m | 0.388±0.034 | 0.627±0.044 | **0.770±0.052** |
| | 3s5z__vs__3s6z | 0.162±0.066 | 0.012±0.006 | **0.586±0.022** |
| | 5s10z | 0.365±0.090 | 0.226±0.110 | **0.483±0.049** |
| | MMM2 | 0.350±0.065 | 0.816±0.050 | **0.959±0.012** |
| GRF | 3_vs_1_WK | 0.097±0.028 | 0.657±0.031 | **0.736±0.127** |
| | CA__Easy | 0.017±0.007 | 0.148±0.045 | **0.276±0.051** |
| | CA__Hard | 0.009±0.006 | **0.101±0.053** | 0.023±0.010 |

A more accurate individual value estimation determines the effectiveness of ECA. While obtaining ground-truth state values in complex environments is infeasible, the results show that the credit assignment of our method offers effective guidance for determining agent contributions. The agent credits computed based on past experiences can efficiently correct individual values in similar states, establishing effective knowledge transfer and minimizing the gap between current and optimal policies.

ECA produces an episodic memory table at each run involving 200,000 to 400,000 abstract states and the corresponding agent credits. The construction of episodic memory takes only a few minutes, which is negligible compared to the cost of MARL policy optimization. Based on the experimental performance and episodic memory size, ECA effectively and efficiently handles high-dimensional MARL environments.

### 4.3 Evaluation of ECA on different credit computation methods

ECA utilizes historical experiences to improve credit assignment, thus enhancing the accuracy of individual and joint value estimations. Multiple alternatives to agent credit calculation can be considered. We compare our proposed method with other credit calculation methods that span different levels of detail when evaluating the impact of the agent's performance on the team. Reward confidence scores of global states (*i.e.*, *state__rcs*) measure the expected episodic returns of team cooperation without highlighting individual agent or action impacts (see Equation 6 in Section 2.3). Reward confidence scores of global state and local action pairs as defined in Equation 8 (*i.e.*, *state__action__rcs*) quantify the influence of individual actions on team episodic returns as defined in Equation 5. In contrast, *agent__credit* (our proposed method) outlined in Equation 9 measures an agent's contribution by considering all feasible actions in a state rather than focusing on individual actions.

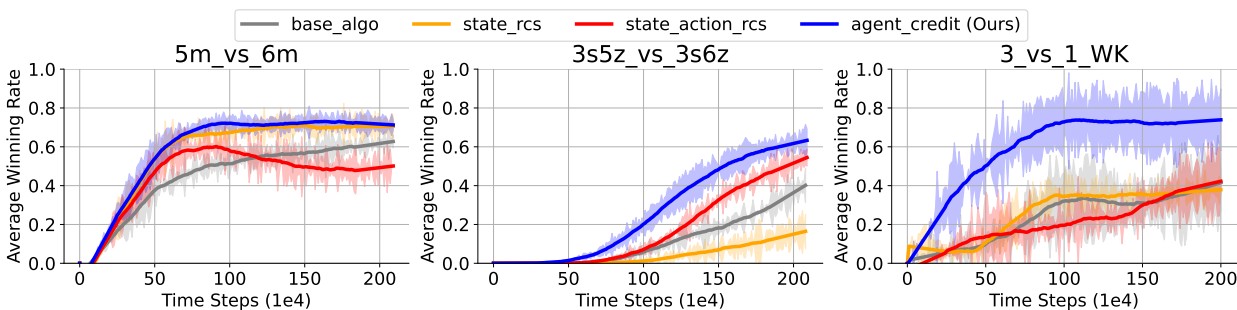

Figure 4: The winning rates of learned policies by incorporating different intrinsic rewards with the individual value estimation. Note that in *base__algo*, we do not perform credit assignment.

Figure 4 depicts the evaluation results of using different agent credits as intrinsic rewards. The results demonstrate that using agent credits as intrinsic rewards to revise the individual estimation significantly outperforms employing global intrinsic rewards, particularly on challenging tasks. Moreover, our method

achieves higher winning rates than directly utilizing individual action reward confidence scores as intrinsic rewards. This indicates that our credits provide a more comprehensive and accurate value estimation for agents, whereas using action reward confidence scores might introduce variances due to their sparsity. In conclusion, directly assigning credits to agents and calculating individual credits is a better way for ECA.

## 4.4 Evaluation of ECA on various value integration methods

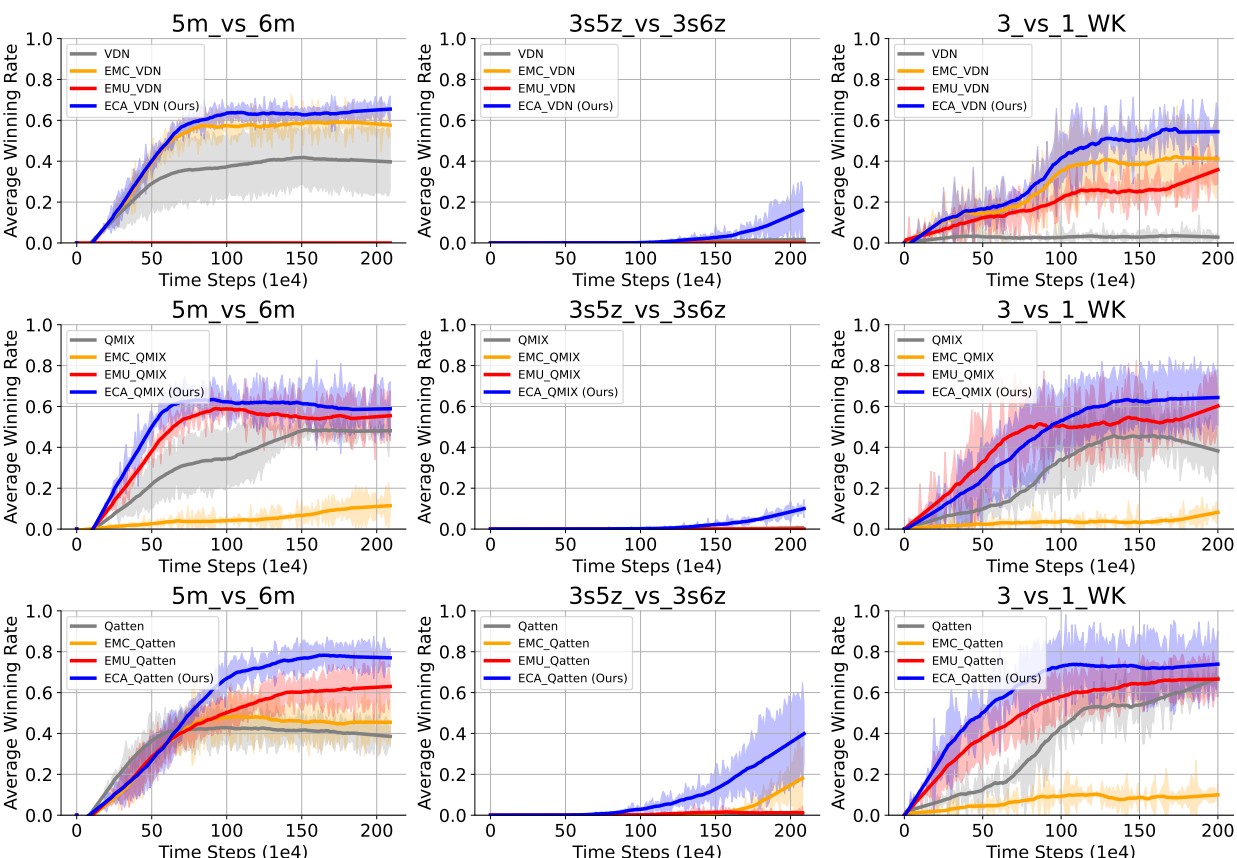

Figure 5: The winning rates of ECA across different value integration methods.

Another strength of ECA is that it can be adapted to different value integration methods. To showcase the adaptability of ECA, we apply it to different base algorithms with diverse value integration methods, including VDN (Sunehag et al., 2017), QMIX (Rashid et al., 2020), and Qatten (Yang et al., 2020a). We also evaluate EMC and EMU on the above value integration methods. Note that for the base algorithms of VDN, QMIX, and Qatten, we do not perform specific explicit reward shaping and credit assignment.

The results depicted in Figure 5 highlight a significant enhancement in sample efficiency when ECA is incorporated with these base algorithms. Such results indicate that ECA can be effortlessly integrated into existing MARL pipelines without requiring extensive modifications. Therefore, ECA is a versatile and practical method for sample efficient MARL. Moreover, the observed improvements in winning rates across various base algorithms also validate the accuracy of individual contribution credit measurements facilitated by ECA.

Moreover, we found that although EMC and EMU show significant performance improvement in most experiments with different integration methods, ECA outperforms them on all the tasks and establishes the most performance improvement against the base algorithms. Such results further prove the effectiveness of ECA in applying different multi-agent tasks and the potential of combining them with various base algorithms.

# 5 Related Work

## 5.1 Credit Assignment

Credit assignment in multi-agent settings involves attributing a global and shared reward from the environment to the individual actions of agents, ensuring that each agent's credit accurately reflects their contribution to coordinated performance. Solving inaccurate credit assignment problems has two directions, implicit and explicit credit assignment.

Implicit credit assignment aims to learn associations between agents and the observed rewards without explicitly attributing rewards to specific actions. Existing works have proposed novel implicit credit assignment structures. MADDPG (Lowe et al., 2017) learns individual policies by ascending value gradients. Value Decomposition Networks (VDN) (Sunehag et al., 2017) aggregates individual estimations to form the joint action value, while QMIX (Rashid et al., 2020) utilizes a mixer network to coordinate and combine individual values. Qatten (Yang et al., 2020a) incorporates an attention mechanism to capture latent coordination relationships among agents, and Qplex (Wang et al., 2021a) employs a dueling network to circumvent direct optimization from monotonic assumptions. DOP (Wang et al., 2021b) decomposes the central critic into weighted individuals. FACMAC (Peng et al., 2021) employs a centralized gradient estimator for credit assignment. However, implicit credit assignment may be limited by constraints such as monotonicity (Rashid et al., 2020).

Unlike implicit methods, explicit methods directly assign credits to agents. Difference rewards (DR) has been employed as an explicit method to enable agents to learn from a shaped reward by comparing the global reward to the reward obtained when an agent's action is replaced with a default action(Wolpert & Tumer, 2001; Tumer & Agogino, 2007; Proper & Tumer, 2012; Foerster et al., 2018; Tumer et al., 2002). However, DR could be more efficient due to the necessity for separate estimations of baselines and its diminished effectiveness in the presence of complex cooperative behaviors. Differently, COMA(Foerster et al., 2018) adopts agent-specific advantage functions to compute the agent credits, and Shapley Q-value is used to evaluate the possible expectation scale of available actions. Jiang *et al.* factorize and reshape team rewards into agent-specific rewards, thus accurately approximating heterogeneous agent contributions and improving energy efficiency(Jiang et al., 2023). However, learning accurate agent credit is still challenging due to the complex environments and agent interactions. Different from existing work, ECA focuses on agent-level explicit credit assignment and utilizes past interaction experiences to estimate the current credits of agents.

## 5.2 Neural Episodic Control

Inspired by the hippocampus mechanism observed in mammals, episodic control-based methods efficiently store and leverage past similar experiences to process new tasks(Lengyel & Dayan, 2007). These methods have found application in model-free Deep Reinforcement Learning (DRL) tasks, particularly for retrieving episodic memory-based state values(Blundell et al., 2016; Pritzel et al., 2017) to address sample inefficiency. Developing episodic control-based methods encompasses various aspects, including state clustering, storage structure, and memory usage. In the realm of state clustering methods, prior approaches typically employ distance-based measurements, such as $k$-nearest neighbors (KNN)(Kuznetsov & Filchenkov, 2021), model-based state representation(Le et al., 2021), and exact-matching methods (Lin et al., 2018; Li et al., 2023). Concerning state storage structure, in addition to non-parametric memory (Zhu* et al., 2020; Zhang et al., 2019; Pinto, 2020), Hu et al. (2021) proposed a generalized episodic memory with neural networks. Regarding memory usage, most existing methods combine memory values with a joint objective function (Hansen et al., 2018), while other works focus on utilizing memory values as intrinsic rewards (Li et al., 2023; Na et al., 2024; Zheng et al., 2021). In MARL domains, EMC Zheng et al. (2021) proposed an episodic memory module to record the novelty of states and utilized the novelty measurement as intrinsic rewards to augment policy optimization. EMU Na et al. (2024) used an episodic control module to reward the desired experiences and accelerate policy optimization on MARL tasks. In this paper, ECA adopts an episodic control framework for MARL tasks to compute agent credits and improve the credit assignment of existing MARL algorithms.

Differently, we employ the episodic control mechanism to rectify the credit assignment in MARL algorithms.

# 6 Limitation and Discussion

While ECA significantly enhances the sample efficiency of existing cooperative MARL algorithms, we acknowledge that there are potential concerns and limitations that need to be considered.

By basing the computation of individual credits on global team returns, ECA effectively addresses concerns about potential non-stationarity. This adaptability ensures the resilience of the learning process in dynamic and non-stationary environments, fostering optimism about ECA's potential. Nevertheless, ECA does not guarantee convergence to the ground truth reward. Moreover, ECA might alter the optimal policy since the agent credits are directly added to the individual action values. Addressing this limitation and exploring the theoretical foundations for convergence could be a direction for future work.

In CA_Hard, although EMU performs better than ECA, they all achieve relatively low winning rates. We investigated the reason and found that, due to the difficulty of CA_Hard, there are no highly rewarded experiences to be added to the episodic memory. Thus, memory-based methods, such as EMC, EMU, and ECA, make it hard to generalize previous good experiences. This can be a limitation to ECA. To resolve this problem, we consider that more effective exploration techniques need to be integrated into ECA since the probability of highly rewarded experiences is improved.

ECA builds upon the CTDE framework, which aims to resolve cooperative multi-agent tasks. ECA might not perform well on tasks that CTDE cannot handle properly, such as decentralized training and decentralized execution (DTDE) framework (Jiang & Lu, 2022; Tan, 1993; Tampuu et al., 2017; Palmer, 2020). However, the problems and algorithm paradigm that ECA targets and studies are currently crucial in the cooperative MARL field, making ECA a promising method.

# 7 Conclusion and Future Work

In conclusion, this paper introduces ECA, a simple episodic control-based method for sample-efficient MARL policy optimization. ECA improves the sample efficiency of existing MARL algorithms by directly measuring the agent contribution to the collective success, using the measurement to correct individual action values, and efficiently minimizing the value estimation gap between current and optimal policies. Experimental evaluations conducted on diverse SMAC and GRF tasks have demonstrated the efficacy of ECA. Comparisons with state-of-the-art MARL algorithms and episodic control-based methods have validated the effectiveness of our credit assignment mechanism.

Our future research directions involve (1) improving the scalability and adaptability of ECA to more complex and dynamic environments, (2) applying ECA to more real-world multi-agent scenarios, (3) exploring more promising credit assignment methods to promote the large-scale deployment of existing MARL algorithms, and (4) targeting on safe and reliable MARL algorithms with more novel state measurements.

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

## A    Appendix

### A.1    Evaluation of Hyperparameter in ECA

We conduct experiments on two key parameters: scaling factor $\epsilon$ and grid number N. The scaling factor $\epsilon$ controls how the credit affects individual action value, and N determines the number of intervals for dividing each dimension of the states. The values for $\epsilon$ were selected as follows: 0.0001, 0.0005, 0.001, 0.005, and 0.01. For the grid number N, we tested configurations of 3, 5, and 7. And these experiments are performed in the 3s5z_vs_3s6z environment.

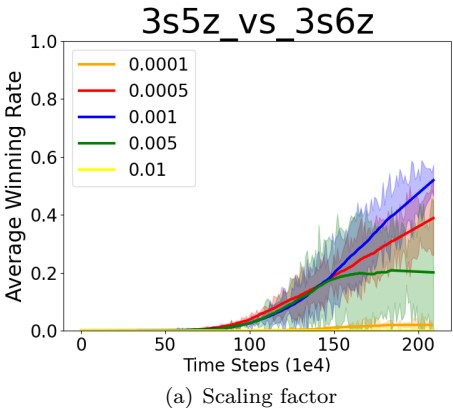
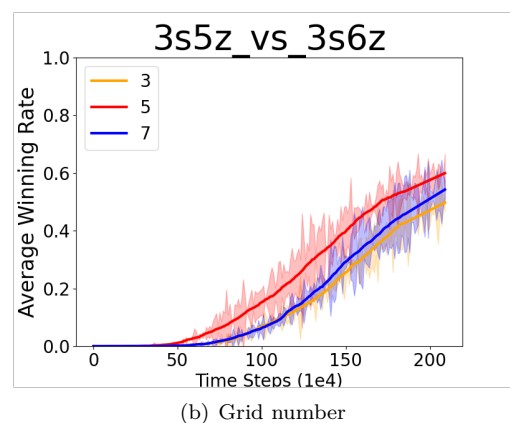

(a) Scaling factor                                   (b) Grid number

Figure 6: Evaluation of Hyperparameter in ECA

Figure 6(a) and figure 6(b) are the experiment results, indicating that using $\epsilon = 0.001$ and N=5 yields the best winning rates on 3s5z_vs_3s6z. For $\epsilon =$, values greater than 0.001 might prioritize the intrinsic rewards excessively and overwhelm the effectiveness of environmental rewards. Values smaller than 0.001 might cause the intrinsic rewards to be ineffective. For grid number, which relates to the granularity of state abstraction, a grid number less than 5 might group more concrete states into the same abstract state, thus limiting the accuracy of state abstraction. On the other hand, a grid number greater than 5 might make the state abstraction too sparse, which affects the accuracy of the statistic of episodic return on each abstract state. Based on the above studies, setting $\epsilon$ to 0.01 and grid number N to 5 is the best choice for the selected tasks.

### A.2    Evaluation of ECA on the Matrix Game

We evaluate ECA on a pathological matrix game proposed by Yang et al. (2020b), as shown in the left alignment of Figure 7. The matrix figure describes the state transition logic and the tabular reward functions. In the game, two agents collaborate to receive rewards under three state categories. The game ends when two agents get 0 rewards. The two agents can earn 13 returns at the most, only if they keep select action

(0, 0) at the initial and intermediate states but act (1, 1) at the final states. The goal is to learn a policy to follow such an optimal path. We experiment with ECA on this game since it has fixed payoff metrics, which makes it convenient to observe the differences between the optimal and our policy. The payoff metrics give the reward value in each state. Note that ECA is implemented based on QMIX (Rashid et al., 2020) in this experiment.

The right alignment of Figure 7 shows the results, indicating that ECA converges to higher episodic returns than QMIX. Moreover, ECA is more likely to act by the optimal path than QMIX, *i.e.*, earn 13 returns. We note that in QMIX, the number of episodes achieving 10 returns is significantly fewer than those earning 13 returns before policy convergence, which suggests that QMIX is more likely to converge to a sub-optimal policy, as it requires more optimal-path samples with 13 returns to update the value functions and influence action selection. In contrast, ECA has an episodic memory module, it can compute and store agent credits in episodic memory upon even a one-time occurrence of optimal-path samples and continuously perform reward shaping to enforce updates towards the optimal policy. Therefore, despite the effectiveness of existing credit assignments in QMIX, ECA performs a more direct and efficient credit assignment to the team.

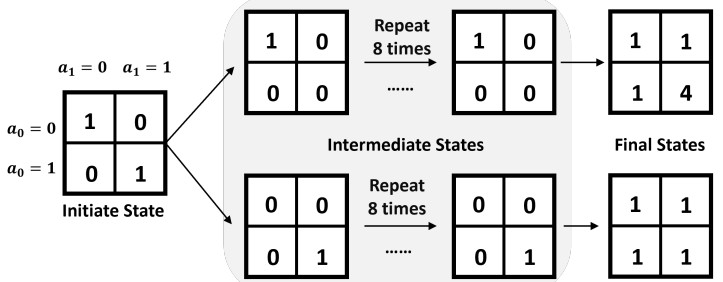 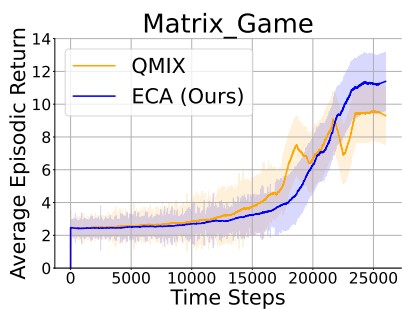

Figure 7: The left align is the state transition logic and tabular reward functions of the matrix game Yang et al. (2020b), where $a_i$ is the action of $i^{th}$ agent. The right align shows trends of the episodic returns during policy optimization.

## A.3 Evaluation of Existing Credit Assignment Methods

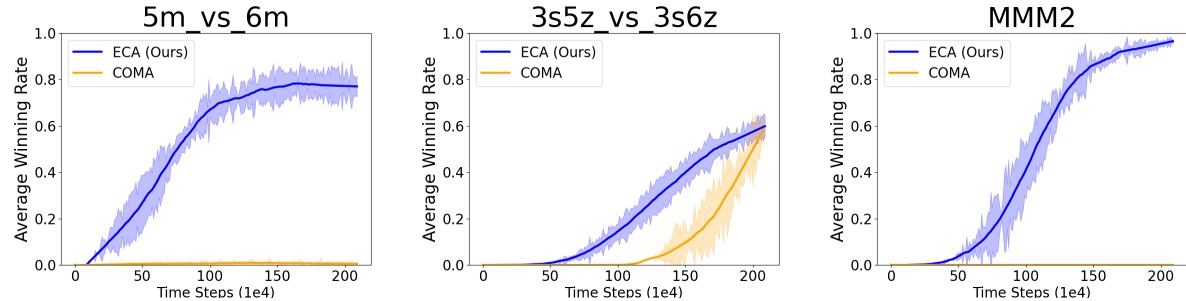

Figure 8: Experimental results of ECA and COMA on 5m_vs_6m, 3s5z_vs_3s6z, and MMM2.

We evaluate existing credit assignment works, such as COMA Foerster et al. (2018), to 5m_vs_6m, 3s5z_vs_3s6z, and MMM2. We select these tasks since the policy needs to control more agents, and the task is more difficult than relative tasks, such as 3s_vs_5z. The implementation of COMA comes from an open-source repository: `https://github.com/jk96491/SMAC`. Within 2M environmental steps and three times repeat on each task, we find that the policies learned by COMA achieve similar winning rates as ECA on 3s5z_vs_3s6z. The trends show that COMA is not converged, and it is promising to achieve higher performance. However, COMA struggles on 5m_vs_6m and MMM2, where the winning rates of learned policies are around 0. Therefore, we infer that ECA is more stable than COMA on the selected tasks.

## A.4 Evaluation of ECA on State Space Exploration

To investigate the influence of ECA on policy exploration, we test ECA on 3s5z_vs_3s6z and MMM2. Specifically, we count the state numbers during training and plot the state number trends on different time steps. Note that concrete state vectors with float components are hard to count, so we use state abstraction in ECA to transform these concrete states into discrete, countable abstract states.

Figure 9 and Figure 10 indicate that ECA converted more than 1.5M and 1.2M abstract states, while Qplex (*i.e.*, the backbone algorithm of ECA) covers fewer abstract states than 1.2M and 0.6M, respectively. Such a result can prove that ECA does not significantly limit the exploration of policies. We infer that the ECA policy achieves superior performance compared to Qplex, thus getting more state space to be explored.

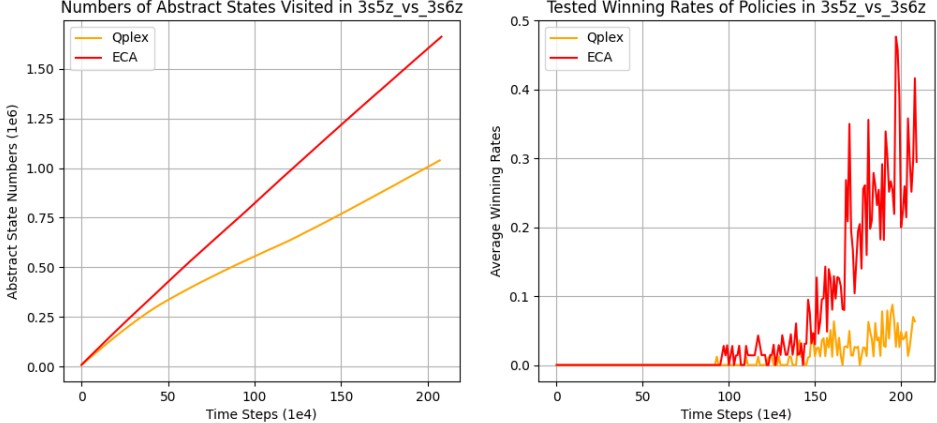

Figure 9: State Space Exploration of ECA on 3s5z_vs_3s6z.

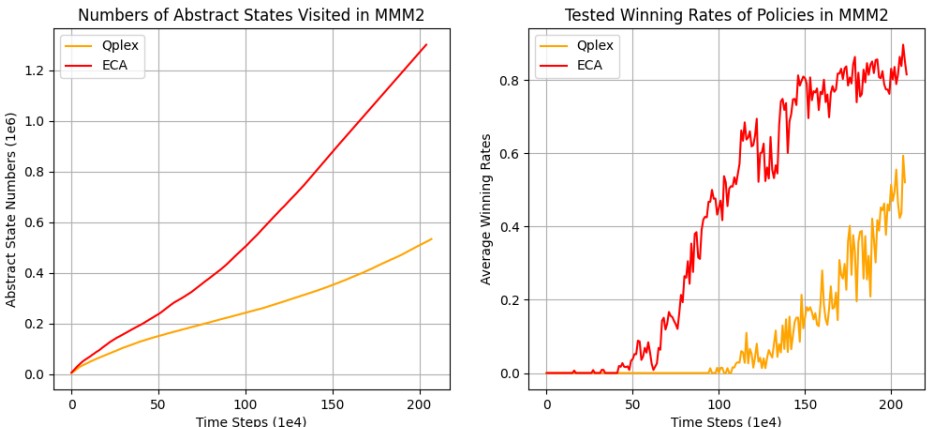

Figure 10: State Space Exploration of ECA on MMM2.

