# OpenReview forum: "Efficient Credit Assignment in Cooperative Multi-Agent Reinforcement Learning"
_TMLR — Rejected by TMLR_

### Review · Reviewer_Hfvc · 2025-04-02

**Summary Of Contributions:**

The paper introduces ECA (Efficient Credit Assignment), an episodic control-based method aimed at enhancing the sample efficiency of cooperative MARL algorithms.  Its key contribution lies in leveraging episodic memory to compute individual agent credits, which are then used as intrinsic rewards to adjust individual action values.

**Audience:**

Yes

**Claims And Evidence:**

Yes

**Requested Changes:**

As discussed in the weaknesses section, a more rigorous experimental section should be provided, including:

- Visualization of explicit credit assignment to support its claims.

- More consistent and reproducible performance results.

- Ablation studies analyzing the impact of key hyperparameters.

**Strengths And Weaknesses:**

Strengths:

- ECA presents a novel approach to credit assignment by utilizing episodic memory to compute individual agent credits.

- The method builds on established concepts such as episodic control and reward shaping, grounding it in a robust theoretical framework.

- Experimental results demonstrate that ECA achieves higher winning rates and faster convergence compared to baselines like EMC and EMU.

- ECA is designed to integrate seamlessly with existing MARL algorithms without requiring significant modifications.

Weakness:

- While credit assignment is a well-known challenge in MARL, the paper would benefit from concrete examples illustrating how improper credit assignment hinders performance, particularly by highlighting the limitations of existing methods.

- Value factorization methods are already known to enhance credit assignment. Since the paper claims to provide explicit credit assignment, it should explicitly demonstrate this improvement—potentially by visualizing credit assignments in benchmark tasks such as SMAC or GRF.

- The reported performance results are inconsistent with the EMU paper. For example, in 5m_vs_6m and MMM2, EMU achieves over 90% win rates within 2M steps, whereas this paper reports only 60% and 80%, respectively. Similar inconsistencies arise in GRF tasks such as ca_easy and ca_hard.

- ECA's performance may be sensitive to hyperparameters, such as the scaling factor ε (set to 0.001) and state abstraction parameters (e.g., N = 5 in state discretization). However, the paper does not provide a thorough sensitivity analysis to assess the impact of these choices.

---

> ### Author Response · Authors · 2025-04-28
> **Response to reviewer Hfvc, weakness 1**
>
> ## [R1 W1] While credit assignment is a well-known challenge in MARL, the paper would benefit from concrete examples illustrating how improper credit assignment hinders performance, particularly by highlighting the limitations of existing methods.
>
> [Matrix_Game](https://anonymous.4open.science/r/ECA_Response/matrix.png)
>
> [Matrix_Result](https://anonymous.4open.science/r/ECA_Response/matrix_result.jpg)
>
> Thanks for advising us on how to illustrate the effectiveness of the credit assignment. We strongly agree with the reviewer and provided a small example to (1) explain how insufficient credit assignment hinders performance and (2) discuss existing work.
>
> We evaluate ECA on a pathological matrix game proposed by Yang [1]. The matrix figure describes the state transition logic and the tabular reward functions. In the game, two agents collaborate to receive rewards under three state categories. The game ends when two agents get 0 rewards. The two agents can earn 13 returns at the most, only if they keep select action $(0,0)$ at the initial and intermediate states but act $(1,1)$ at the final states. The goal is to learn a policy to follow such an optimal path. We experiment with ECA on this game since it has fixed payoff metrics, which makes it convenient to observe the differences between the optimal and our policy. The payoff metrics give the reward value in each state. Note that ECA is implemented based on QMIX [2] in this experiment.
>
> The results show that ECA converges to higher episodic returns than QMIX. Moreover, ECA is more likely to act by the optimal path than QMIX, \ie, earn 13 returns. We note that in QMIX, the number of episodes achieving 10 returns is significantly fewer than those earning 13 returns before policy convergence, which suggests that QMIX is more likely to converge to a sub-optimal policy, as it requires more optimal-path samples with 13 returns to update the value functions and influence action selection. In contrast, ECA has an episodic memory module, it can compute and store agent credits in episodic memory upon even a one-time occurrence of optimal-path samples and continuously perform reward shaping to enforce updates towards the optimal policy. **Therefore, despite the effectiveness of existing credit assignments in QMIX, ECA performs a more direct and efficient credit assignment to the team.**
>
> [1] Yang, Y., Wen, Y., Wang, J., Chen, L., Shao, K., Mguni, D., & Zhang, W. (2020, November). Multi-agent determinantal q-learning. In International Conference on Machine Learning (pp. 10757-10766). PMLR.
> [2] Rashid, T., Samvelyan, M., De Witt, C. S., Farquhar, G., Foerster, J., & Whiteson, S. (2020). Monotonic value function factorisation for deep multi-agent reinforcement learning. Journal of Machine Learning Research, 21(178), 1-51.

---

> ### Author Response · Authors · 2025-04-28
> **Response to reviewer Hfvc, weakness 2**
>
> ## [R1 W2] Value factorization methods are already known to enhance credit assignment. Since the paper claims to provide explicit credit assignment, it should explicitly demonstrate this improvement—potentially by visualizing credit assignments in benchmark tasks such as SMAC or GRF.
>
> Thanks for the advice. Take 3s5z_vs_3s6z as an example; we collected episodes and conducted a visualization analysis of our agent credits. We find that credits are relatively high when the agent attacks and decreases enemy health points, and the team has an advantage and wins the game, and the credits are low when the agent is attacked and the team loses the game. Moreover, the agent credits tend to be consistent in the middle and late stages; we infer that the battle results (i.e., win or lose) are already predictable due to past experiences.
>
> [episode1](https://anonymous.4open.science/r/ECA_Response/episode1.png)
>
> [step7](https://anonymous.4open.science/r/ECA_Response/episode1/image_7.png) [step8](https://anonymous.4open.science/r/ECA_Response/episode1/image_8.png) [step9](https://anonymous.4open.science/r/ECA_Response/episode1/image_9.png) [step10](https://anonymous.4open.science/r/ECA_Response/episode1/image_10.png) [step11](https://anonymous.4open.science/r/ECA_Response/episode1/image_11.png)
>
> In episode 1, the team killed the first enemy at step 11 and gained an advantage in agent numbers at step 20. Considering the team has 3s+5z=8 agents while the enemy has 9 agents, we can confirm that the team established an advantage very early. Note that although the first enemy died at step 11, it was attacked earlier. Our agent credits are also high around step 8 when the eliminated enemy is tracked by our agent skills (a black symbol inside the enemy agent icon).
>
> [episode2](https://anonymous.4open.science/r/ECA_Response/episode1.png)
>
> [step3](https://anonymous.4open.science/r/ECA_Response/episode2/image_3.png) [step4](https://anonymous.4open.science/r/ECA_Response/episode2/image_4.png)
>
> For example, in episode 2, we can see that in step 3, agent 5_z is assigned with very low credit since it is attacked from step 4. Moreover, the credits of all the agents are low since the enemies are encompassing and attacking the team.
>
> Moreover, we collected 78 episodes, and in each episode, we conducted a statistic of correlation of the **average agent credits** and the **gaps between average team health points and enemy health points**. We find that 63.29% of all the episodes suggest a positive correlation with a p-value less than 0.01. This result shows that our agent credits are closely related to team and enemy health points. The reason is that the agent credits in ECA are computed based on the episodic returns, and the team needs to eliminate all the enemies and earn the final reward (the only reward equal to the episodic return). For the episodes that suggest a negative correlation, we find that their p-value is usually greater than 0.01, which means that the negative correlation is insignificant. We infer that a more complex analysis is required to understand the meaning of agent credit computation.

---

> ### Author Response · Authors · 2025-04-28
> **Response to reviewer Hfvc, weakness 3**
>
> ## [R1 W3] The reported performance results are inconsistent with the EMU paper. For example, in 5m_vs_6m and MMM2, EMU achieves over 90% win rates within 2M steps, whereas this paper reports only 60% and 80%, respectively. Similar inconsistencies arise in GRF tasks such as ca_easy and ca_hard.
>
> Thanks for pointing out this problem. We have investigated the performance of EMU on 5m_vs_6m, MMM2, CA_Easy, and CA_Hard. We ran 2M steps on each task and repeated them 3 times to counteract the randomness. We find that the results of EMU are not stable. Specifically, the average win rates and their corresponding standard deviations are as follows:
>
> [5m_vs_6m](https://anonymous.4open.science/r/ECA_Response/EMU-5m_vs_6m.png)
> - 5m_vs_6m: 0.627±0.044 → 0.787±0.034; (Increased)
>
> [MMM2](https://anonymous.4open.science/r/ECA_Response/EMU-MMM2.png)
> - MMM2: 0.816±0.050→0.728±0.109; (Decreased)
>
> [CA_Easy](https://anonymous.4open.science/r/ECA_Response/EMU-CA_Easy.png)
> - CA_Easy 0.148±0.045→0.315±0,063;(Increased)
>
> [CA_Hard](https://anonymous.4open.science/r/ECA_Response/EMU-CA_Hard.png)
> - CA_Hard 0.101±0.053→0.005±0.008.(Decreased)
>
> Such a result reminds us that repeating is not enough for EMU. Therefore, we will continuously experiment with EMU on the tasks above and the rest. We will update the paper once we have the stable results.

---

> ### Author Response · Authors · 2025-04-28
> **Response to reviewer Hfvc, weakness 4**
>
> ## [R1 W4] ECA's performance may be sensitive to hyperparameters, such as the scaling factor ε (set to 0.001) and state abstraction parameters (e.g., N = 5 in state discretization). However, the paper does not provide a thorough sensitivity analysis to assess the impact of these choices.
>
> Thanks for this question. We agree with the reviewer that the sensitivity of hyperparameters requires detailed studies. We conduct this study on scaling factor epsilon and grid number in ECA. The epsilon is set to 0.0001, 0.0005, 0.001, 0.005, and 0.01 for comparison. The grid number is set to 3, 5, and 7 for comparison. We experimented with the hyperparameters on 3s5z_vs_3s6z. The results are as follows:
>
> [scaling_factor](https://anonymous.4open.science/r/ECA_Response/scaling_factor-3s5z_vs_3s6z.jpg)
>
> [grid_num](https://anonymous.4open.science/r/ECA_Response/grimnum-3s5z_vs_3s6z.png)
>
>
> The results show that using 0.001 as the epsilon and 5 as the grid number of ECA, the best winning rates are 3s5z_vs_3s6z. For epsilon, a greater value than 0.001 might weigh more on the intrinsic rewards and overwhelm the effectiveness of environmental rewards. A smaller value than 0.001 might cause the intrinsic rewards to be ineffective. For grid number, which relates to the granularity of state abstraction, a smaller grid number than 5 might group more concrete states into the same abstract state, thus limiting the accuracy of state abstraction. Moreover, a greater grid number than 5 might make the state abstraction too sparse, which affects the accuracy of the statistic of episodic return on each abstract state. Based on the above studies, setting epsilon and grid number to 0.001 and 5 is the best choice for the selected tasks.
>
>
>
> ## [R1 Requested Changes:]
> #### As discussed in the weaknesses section, a more rigorous experimental section should be provided, including:
> - Visualization of explicit credit assignment to support its claims.
> - More consistent and reproducible performance results.
> - Ablation studies analyzing the impact of key hyperparameter.
>
> We thank the reviewer's effort and advice very much. We have added the requested changes to the paper and the replies.

---

### Review · Reviewer_B9R9 · 2025-04-08

**Summary Of Contributions:**

This paper provides a novel way of resolving the credit assignment problem in cooperative MARL. Usually in cooperative MARL, all agents receive the same numerical reward at each time step, irrespective of their contributions to the overall objective. This makes it extremely difficult to accurately assign credit to individual agents, which in turn leads to difficulties in learning good policies. In this paper, a new method (ECA) is proposed to relax the credit assignment limitation of cooperative MARL. ECA uses episodic memory, stores and clusters interaction experiences from the memory, which is then used to assign distinct rewards to individual agents on the basis of their overall contributions to achieving the joint/team objective for all agents in this cooperative environment. ECA functions in two phases: episodic memory construction and episodic control. In the first phase, global states are transformed and grouped into a set of abstract states. Agent experience trajectories with the abstract states are used to compute episodic returns (assigned as rewards/credits) for individual agents. In the second phase, policy optimization is used to further correct the individual rewards. Multiple experiments on standard cooperative MARL testbeds (SMAC and GRF) showcase the advantages of ECA as compared to a set of recent baselines.

**Audience:**

Yes

**Claims And Evidence:**

No

**Requested Changes:**

Please address the weaknesses listed in the previous section.

In addition, the entire paper requires a thorough proofreading for the purposes of resolving typos, correcting grammar and organization. There are several typos and awkward phrasings that need to be addressed in the paper. I am including some of them from the beginning and end of the paper here, but it is simply impossible to mention everything.

Page 2: "limitation of monotonic that" -> "limitation of monotonic improvement that"? (awkward, seems to miss a word).

Page 2: "complex environments" -> "complexity of the environments"

Page 7" "that varied by" -> "that varies by"

Page 11: "Difference rewards have" -> "Difference rewards has"

Page 11: "tasks. we" -> "tasks. We"

Page 11: "Hu et al. (2021). proposed" -> "Hu et al. (2021) proposed"

Page 11: "tasks suitable for such tasks" -> awkward, consider rephrasing.

Page 12: "targeting on safe" -> "targeting safe"

**Strengths And Weaknesses:**

Strengths:

The paper chooses a nice, timely research direction. The problem of credit assignment in cooperative MARL indeed requires innovative solutions, and this paper makes a solid attempt for providing the same. The experiments appear comprehensive.


Weaknesses:

The writing and presentation of the paper needs improvement. I have several confusions about the paper, which I am listing here:

1. Why is the State Abstraction technique provided as part of the Section on Preliminaries? Is this not a contribution of this paper? If not, can you provide citations to previous work that provides this technique?

2. I am unable to understand the details of the state abstraction. An example would be very helpful to see what the different variables mean, how these are calculated, and why it is effective.

3. Similarly, it is hard to follow all the details for the Algorithm 1 and 2. The explanation uses a rather poor combination of mathematical symbols and pseudocode descriptions to explain the working of the algorithm. The intuitions of the different design choices are either missing or extremely unclear.  It would be helpful to include a simple example that guides the reader through all the steps.

4. In the Related Work section, the paper lists several closely related baselines, but do not use them for the experimental comparisons. More explanation is necessary for the choice of baselines and reasons for ignoring several baselines mentioned in the related works.

5. There are several awkward and unclear phrasings that hinder the flow of reading. For example, "freshness of states" (what is that?), "amalgamated into joint-action values" (unclear meaning), "trained based on disparity between", etc. The paper requires a thorough proofreading.

6. While all the results show performances using the "Average Winning Rate" as the metric, this does not quite capture the individual agent credit assignment provided by ECA and the qualitative differences of this assignment as compared to the credits used by the baselines. The paper claims that "our results show that accurate agent credit estimation offers effective guidance for determining agent contributions". However, I do not think there is sufficient evidence for this statement. The improved performances provided by ECA could simply be due to deficiencies seen in the baselines, as opposed to actual advantages of ECA. For example, the curiosity based exploration conducted by EMC may be inefficient in the environments considered. It would be helpful to see the credits computed by ECA in at-least a simple grid based environment to understand its strengths.

7. More explanation is needed in the scenarios where ECA's performance is inferior as compared to the baselines. For example, EMU provides better performance than ECA in Figure 3 (CA_Hard). What weaknesses of ECA are exposed in these environments? Does ECA fail in comparatively harder settings for some reason?

---

> ### Author Response · Authors · 2025-04-28
> **Response to reviewer B9R9, weakness 1, 2, and 3**
>
> ## [R2 W1] Why is the State Abstraction technique provided as part of the Section on Preliminaries? Is this not a contribution of this paper? If not, can you provide citations to previous work that provides this technique?
>
> Thanks for this question. We provide state abstraction as a part of the Section on Preliminaries since it is a technique of a published paper, NECSA [1]. State abstraction is not an ECA contribution. We provided the citations, Preliminaries, and related work.
>
> ## [R2 W2] I am unable to understand the details of the state abstraction. An example would be very helpful to see what the different variables mean, how these are calculated, and why it is effective.
>
> Thanks for this advice. To better explain the idea of state abstraction, we provided a simple example in the last paragraph of 2.2 State Abstraction.
>
> "Given a two-dimensional concrete state (0.32, 0.73), each feature ranges from 0 to 1. We set interval N = 5, and then
> each dimension is split into five intervals: $\eta_1$ = [0, 0.2), $\eta_2$ = [0.2, 0.4), $\eta_3$ = [0.4, 0.6), $\eta_4$ = [0.6, 0.8), $\eta_5$ =
> [0.8, 1.0] can be represented as 1, 2, 3, 4, and 5, respectively. 0.32 falls in $\eta_2$, 0.73 falls in $\eta_4$. Thus, the original state (0.32, 0.73) can be represented in an abstract
> state (2, 4). The abstract state is easier to count based on the occurrences than concrete states."
>
> We can hardly find two equal concrete states in high-dimensional RL tasks. Therefore, concrete states consist of float numbers. State abstraction converts the concrete states into abstracted (or discrete) ones. Abstracted states are easy to count. We can compute each abstract state's occurrences and episodic returns based on counting. We count ECA's abstract state and individual action pair occurrences and episodic returns. Thus, each pair's average episodic return (i.e., reward confidence score) can be used to measure the impact of the action on results. Finally, for each agent, we use the average reward confidence scores of all the actions to represent the agent's credit in the state. Overall, the effectiveness of state abstraction comes from converting the concrete states into countable ones.
>
>
> ## [R2 W3] Similarly, it is hard to follow all the details for the Algorithm 1 and 2. The explanation uses a rather poor combination of mathematical symbols and pseudocode descriptions to explain the working of the algorithm. The intuitions of the different design choices are either missing or extremely unclear. It would be helpful to include a simple example that guides the reader through all the steps.
>
> [credit](https://anonymous.4open.science/r/ECA_Response/credits.png)
>
> Thanks for this advice. Adding an example to explain the computation of agent credits helps understand ECA. We added a picture to explain the computation process.
>
> As shown in this figure, historical interactions are recorded, including 3 episodes with their corresponding episodic return, 5 states and 2 agents. Each agent has 2 optional actions: 0 and 1. For example, in episode_2, at state_5, agent_0 and agent_1 both take action 0. The episodic return of episode_2 is 5.
>
> Suppose episode_3, where at state_2, agent_0 takes actions 0. This forms a state-action pair for agent_0: (state_2, 0). We can see that this pair appeared twice, the other one appeared in episode_2. The episodic return for episode_2 is 5, and for episode_3 it is 10. Therefore, the reward confidence score of agent_0 at (state_2, 0) is calculated as: (5+10)/2=7.5.
>
> When calculating the individual credit, all occurred actions must be considered. Since there are two actions (0 and 1), we also need to compute the reward confidence score of agent_0 at (state_2, 1), that is 5 (appeared only once in episode_1). Thus the individual credit of agent_1 at stage_2 is calculated as the average of the reward confidence scores over all occurred actions: (7.5 + 5)/2 = 6.25. And this individual credit 6.25 is then used to  revise the individual action values during training.
>
> In ECA, the state-action pairs and their occurrence counts, corresponding episodic returns are recorded in the episodic memory structure.

---

> ### Author Response · Authors · 2025-04-28
> **Response to reviewer B9R9, weakness 4 and 5**
>
> ## [R2 W4] In the Related Work section, the paper lists several closely related baselines, but do not use them for the experimental comparisons. More explanation is necessary for the choice of baselines and reasons for ignoring several baselines mentioned in the related works.
>
> Thanks for pointing this out. We admit that the comparison is insufficient. We select COMA [2] as a baseline. We compared ECA to COMA on 5m_vs_6m, 3s5z_vs_3s6z, and MMM2. The results are as follows:
>
> [COMA-5m_vs_6m](https://anonymous.4open.science/r/ECA_Response/COMA-5m_vs_6m.png)
> [COMA-3s5z_vs_3s6z](https://anonymous.4open.science/r/ECA_Response/COMA-3s5z_vs_3s6z.png)
> [COMA-MMM2](https://anonymous.4open.science/r/ECA_Response/COMA-MMM2.png)
>
> - We select these tasks since the policy needs to control more agents, and the task is more difficult than relative tasks, such as 3s_vs_5z. The implementation of COMA comes from an [open-source repository](https://github.com/jk96491/SMAC). Within 2M environmental steps and three times repeat on each task, we find that the policies learned by COMA achieve similar winning rates as ECA on 3s5z_vs_3s6z. The trends show that COMA is not converged, and it is promising to achieve higher performance. However, COMA struggles on 5m_vs_6m and MMM2, where the winning rates of learned policies are around 0. Therefore, we infer that ECA is more stable than COMA on the selected tasks.
>
> - For other related works, we do not select them as baselines since they were thoroughly compared by previous works or lack of public implementation [3]. Moreover, we will continuously experiment with COMA on more tasks and are open to adding baselines for comparison if the reviewers require it.
>
> ## [R2 W5] There are several awkward and unclear phrasings that hinder the flow of reading. For example, "freshness of states" (what is that?), "amalgamated into joint-action values" (unclear meaning), "trained based on disparity between", etc. The paper requires a thorough proofreading.
>
> Thanks for the detailed check on our paper. We appreciate this and apologize for such mistakes. We have thoroughly checked the paper. The updated context is highlighted in blue.

---

> ### Author Response · Authors · 2025-04-28
> **Response to reviewer B9R9, weakness 6 and 7**
>
> ## [R2 W6] While all the results show performances using the "Average Winning Rate" as the metric, this does not quite capture the individual agent credit assignment provided by ECA and the qualitative differences of this assignment as compared to the credits used by the baselines. The paper claims that "our results show that accurate agent credit estimation offers effective guidance for determining agent contributions". However, I do not think there is sufficient evidence for this statement. The improved performances provided by ECA could simply be due to deficiencies seen in the baselines, as opposed to actual advantages of ECA. For example, the curiosity based exploration conducted by EMC may be inefficient in the environments considered. It would be helpful to see the credits computed by ECA in at-least a simple grid based environment to understand its strengths.
>
> In cooperative multi-agent reinforcement learning, all the agents work together for team success, and it aims to learn good behaviors through trial-and-error interactions with each other and the environment. Thus, we choose the average winning rate as a metric. However, as the reviewer pointed out, our results cannot prove that the effectiveness of ECA is from "accurate agent credit estimation." We changed the expression "our results show that accurate agent credit estimation offers effective guidance for determining agent contributions." to " the results show that credit assignment of our method offers effective guidance for determining agent contributions." Thanks again for pointing out this important problem.
>
> Moreover, we provide a grid game as an example to explain the effectiveness of ECA. We evaluate ECA on a pathological matrix game proposed by Yang [4]. The matrix figure describes the state transition logic and the tabular reward functions. In the game, two agents collaborate to receive rewards under three state categories. The game ends when two agents get 0 rewards. The two agents can earn 13 returns at the most, only if they keep select action $(0,0)$ at the initial and intermediate states but act $(1,1)$ at the final states. The goal is to learn a policy to follow such an optimal path. We experiment with ECA on this game since it has fixed payoff metrics, which makes it convenient to observe the differences between the optimal and our policy. The payoff metrics give the reward value in each state. Note that ECA is implemented based on QMIX [5] in this experiment.
>
> The results show that ECA converges to higher episodic returns than QMIX. Moreover, ECA is more likely to act by the optimal path than QMIX, \ie, earn 13 returns. We note that in QMIX, the number of episodes achieving 10 returns is significantly fewer than those earning 13 returns before policy convergence, which suggests that QMIX is more likely to converge to a sub-optimal policy, as it requires more optimal-path samples with 13 returns to update the value functions and influence action selection. In contrast, ECA has an episodic memory module. It can compute and store agent credits in episodic memory upon even a one-time occurrence of optimal-path samples and continuously perform reward shaping to enforce updates towards the optimal policy. **Therefore, despite the effectiveness of existing credit assignments in QMIX, ECA performs a more direct and efficient credit assignment to the team.**
>
> [Matrix_Game](https://anonymous.4open.science/r/ECA_Response/matrix.png)
>
> [Matrix_Result](https://anonymous.4open.science/r/ECA_Response/matrix_result.jpg)
>
>
>
> ## [R2 W7] More explanation is needed in the scenarios where ECA's performance is inferior as compared to the baselines. For example, EMU provides better performance than ECA in Figure 3 (CA_Hard). What weaknesses of ECA are exposed in these environments? Does ECA fail in comparatively harder settings for some reason?
>
> As we replied in [R2 W6], we consider ECA's effectiveness from a more direct and efficient credit assignment than previous works. However, ECA computes credit assignments based on past experiences. ECA can repeatedly utilize past highly rewarded experiences, even if they occur just a few times. Instead, previous works like QMIX might need more samples to generalize the highly-rewarded experiences.
>
> In CA_Hard, although EMU performs better than ECA, they all achieve relatively low winning rates. We investigated the reason and found that, due to the difficulty of CA_Hard, there are no highly rewarded experiences to be added to the episodic memory. Thus, memory-based methods, such as EMC, EMU, and ECA, make it hard to generalize previous good experiences. This can be a limitation to ECA. To resolve this problem, we consider that more effective exploration techniques need to be integrated into ECA since the probability of highly rewarded experiences is improved. We will add the above concerns and discuss them in future work.

---

> ### Author Response · Authors · 2025-04-28
> **Response to reviewer B9R9, requested changes and references**
>
> ## [R2 Requested Changes:]
>
> - Please address the weaknesses listed in the previous section.
> - In addition, the entire paper requires a thorough proofreading for the purposes of resolving typos, correcting grammar and organization. There are several typos and awkward phrasings that need to be addressed in the paper. I am including some of them from the beginning and end of the paper here, but it is simply impossible to mention everything.
>   - Page 2: "limitation of monotonic that" -> "limitation of monotonic improvement that"? (awkward, seems to miss a word).
>   - Page 2: "complex environments" -> "complexity of the environments"
>   - Page 7" "that varied by" -> "that varies by"
>   - Page 11: "Difference rewards have" -> "Difference rewards has"
>   - Page 11: "tasks. we" -> "tasks. We"
>   - Page 11: "Hu et al. (2021). proposed" -> "Hu et al. (2021) proposed"
>   - Page 11: "tasks suitable for such tasks" -> awkward, consider rephrasing.
>   - Page 12: "targeting on safe" -> "targeting safe"
>
>
> We thank the reviewr's effort and advice very much. We have added the requested changes to the paper and the replies.
>
> ## Reference
>
> [1] Li, Z., Zhu, D., Hu, Y., Xie, X., Ma, L., ZHENG, Y., ... & Zhao, J. Neural Episodic Control with State Abstraction. In The Eleventh International Conference on Learning Representations.
>
> [2] Foerster, J., Farquhar, G., Afouras, T., Nardelli, N., & Whiteson, S. (2018, April). Counterfactual multi-agent policy gradients. In Proceedings of the AAAI conference on artificial intelligence (Vol. 32, No. 1).
>
> [3] Jiang, K., Liu, W., Wang, Y., Dong, L., & Sun, C. (2023). Credit assignment in heterogeneous multi-agent reinforcement learning for fully cooperative tasks. Applied Intelligence, 53(23), 29205-29222.
>
> [4] Yang, Y., Wen, Y., Wang, J., Chen, L., Shao, K., Mguni, D., & Zhang, W. (2020, November). Multi-agent determinantal q-learning. In International Conference on Machine Learning (pp. 10757-10766). PMLR.
>
> [5] Rashid, T., Samvelyan, M., De Witt, C. S., Farquhar, G., Foerster, J., & Whiteson, S. (2020). Monotonic value function factorisation for deep multi-agent reinforcement learning. Journal of Machine Learning Research, 21(178), 1-51.

---

### Review · Reviewer_uShT · 2025-04-14

**Summary Of Contributions:**

This paper presents an episodic control-based method to address the credit assignment problem in multi-agent reinforcement learning (MARL). Episodic memory is employed to store and cluster past trajectories, while an intrinsic reward signal enhances individual and joint value estimation. Experimental results on the SMAC and GRF benchmarks validate the effectiveness of the proposed approach.

**Audience:**

Yes

**Claims And Evidence:**

No

**Requested Changes:**

1.	The distinction between EMC and EMU should be emphasized. Section 5.2 currently lacks detailed explanation.

2.	The influence of the intrinsic reward should be systematically analyzed.

3.	As the paper addresses explicit credit assignment, related works introduced in Section 5.1 should be compared more thoroughly.

**Strengths And Weaknesses:**

Strengths

1.	The paper is well-organized, with the two-phase method clearly explained.

2.	Experimental results on two benchmarks demonstrate the proposed method’s effectiveness.

3.	The computational overhead introduced by the episodic memory is minimal.

Weaknesses

1.	The main contribution relies on the "reward confidence score," which prioritizes episodes with higher returns. This may lead to insufficient exploration in MARL scenarios.

2.	Agent contributions are estimated based on past abstract states, making the method highly dependent on the effectiveness of state abstraction.

3.	The intrinsic reward may alter the optimal policy.

4.	[Minor] Typos are present—for example, Section 3.2 on Page 5 ($\mathcal{E}_\psi$).


Questions

1.	Is the clustering of states updated during training?

2.	How is $c_i$ chosen in different scenarios? Has sensitivity analysis been conducted?

3.	Can the proposed method be adapted for policy-based approaches such as MAPPO?

4.	Are other clustering methods suitable for this framework?

---

> ### Author Response · Authors · 2025-04-28
> **Response to Reviewer uShT, weakness 1, 2, 3, and 4**
>
> ## [R3 W1] The main contribution relies on the "reward confidence score," which prioritizes episodes with higher returns. This may lead to insufficient exploration in MARL scenarios.
>
>
> [3s5z_vs_3s6z](https://anonymous.4open.science/r/ECA_Response/3s5z_vs_3s6z_exploration.png)
> [MMM2](https://anonymous.4open.science/r/ECA_Response/MMM2_exploration.png)
>
> Thanks for pointing out this concern of ECA. To investigate the influence of ECA on policy exploration, we conducted a small verification on 3s5z_vs_3s6z and MMM2. Specifically, we count the state numbers during training and plot the state number trends on different time steps. Note that concrete state vectors with float components are hard to count, so we use state abstraction in ECA, which converts concrete states to discrete and countable abstract states. In the results, we find ECA converted more than 1.5M and 1.2M abstract states, while Qplex (i.e., the backbone algorithm of ECA) covers fewer abstract states than 1.2M and 0.6M, respectively. Such a result can prove that ECA does not significantly limit the exploration of policies. We infer that ECA policy learns better performance than Qplex, thus getting more state space to be explored.
>
> However, we still agree with the reviewer since ECA focuses more on exploiting past high-return experiences than exploring the state space. We will add this to future work and consider a MARL algorithm that can balance exploitation and exploration.
>
>
> ## [R3 W2] Agent contributions are estimated based on past abstract states, making the method highly dependent on the effectiveness of state abstraction.
>
> We admit that the effectiveness of ECA is highly dependent on ECA. Specifically, we conducted experiments on the grid numbers of grid-based state abstraction. We find that the granularities of state abstraction establish distinct experimental results, making grid number a sensitive and important hyperparameter of ECA.
>
> ## [R3 W3] The intrinsic reward may alter the optimal policy.
>
> We agree with the reviewer since ECA is focusing on applying an episodic memory-based credit assignment mechanism while lacking a theoretical basis. Therefore, we are not sure if ECA alters the optimal policy. However, as introduced in previous works, such as NEC, MFEC, and RND, episodic memory and reward shaping-based methods perform effectively in various RL tasks despite the decision of policy optimality. We will add the above concerns to the limitation of ECA.
>
> ## [R3 W4] [Minor] Typos are present—for example, Section 3.2 on Page 5 ().
>
> Thanks for pointing this out. We are verry sorry for this and we corrected in the paper.

---

> ### Author Response · Authors · 2025-04-28
> **Response to Reviewer uShT, question 1, 2, 3, and 4**
>
> ## [R3 Q1] Is the clustering of states updated during training?
>
> In ECA, state clustering remains fixed throughout training. Initially, we initialize a random matrix for Gaussian Random Projection to reduce the dimensionality of the state vectors (originally over 100 dimensions). This random matrix is not updated afterward. Additionally, we manually define the upper and lower bounds of the projected state vectors (24 dimensions) and uniformly divide each dimension into grids based on a pre-specified grid number. These bounds also remain unchanged during training. As a result, the clustering of states is static, and the abstract states corresponding to concrete state vectors are used to count state occurrences.
>
> ## [R3 Q2] How is ci chosen in different scenarios? Has sensitivity analysis been conducted?
>
> Thanks for this question. c_i represents reward confidence scores, which is the agent credit of agent i. c_i is computed based on the reward confidence scores of each global-state-individual-action pairs. We provide a simple example in Appendix A.5 to explain the computation.
>
> The sensitivity of c_i can be measured by the episilon in Equation 10. we agree with the reviewer that it requires detailed studies. We conduct this study on scaling factor epsilon in ECA. The epsilon is set to 0.0001, 0.0005, 0.001, 0.005, and 0.01 for comparison.
>
> [scaling_factor](https://anonymous.4open.science/r/ECA_Response/scaling_factor-3s5z_vs_3s6z.jpg)
>
>
> The results show that using 0.001 as the epsilon and 5 as the grid number of ECA aches, the best winning rates are 3s5z_vs_3s6z. For epsilon, we infer that a greater value than 0.001 might weigh more on the intrinsic rewards and overwhelm the effectiveness of environmental rewards. A smaller value than 0.001 might cause the intrinsic rewards to be ineffective. Based on the above studies, setting epsilon to 0.001 is the best choice for the selected tasks.
>
> ## [R3 Q3] Can the proposed method be adapted for policy-based approaches such as MAPPO?
>
> Theoretically, as a highly supplementary method, ECA is applicable in many episodic RL tasks since the core of ECA is to compute the agent credits based on the episodic return. However, applying ECA to MAPPO has two challenges. At first, agent credits in ECA are the average of all the global-state-individual-action returns, which requires the action space to be discrete. Therefore, ECA might be helpful with MAPPO on discrete action spaces, while MAPPO is effective on discrete and continuous action spaces. Another challenge is that MAPPO adopts generalized advantage estimation (GAE), advantage normalization, and value clipping. Such a framework might require ECA to revise the reward shaping method and hyperparameters, such as epsilon in Equation 10.
>
> ## [R3 Q4] Are other clustering methods suitable for this framework?
>
> In previous episodic memory-based methods, such as NEC and MFEC, similar states are grouped by k-nearest neighbors (KNN). KNN-based state clustering has proved to be effective based on the experimental results. So, other clustering methods are also effective in ECA. We use grid-based state abstraction since it is efficient in computing, suitable for high-dimensional tasks, and effective in many RL tasks (Cite NECSA). Nevertheless, we are open to trying different clustering methods. We will add this to the future work.
>
> Again, thanks for the kind advice. We appreciate the questions.

---

> ### Author Response · Authors · 2025-04-28
> **Response to Reviewer uShT, requested changes**
>
> ## [R3 Requested Changes:]
>
> The distinction between EMC and EMU should be emphasized. Section 5.2 currently lacks a detailed explanation.
>
> - We have added explanations of distinctions between EMC and EMU in Section 5.2. The context is highlighted in blue.
>
> The influence of the intrinsic reward should be systematically analyzed.
>
> - It is hard to theoretically analyze intrinsic rewards in ECA since this paper focuses on the straightforward utilization of memory-based credit assignments. Nevertheless, we provided a visualization analysis of intrinsic rewards in ECA. Please refer to the [visualization analysis](https://anonymous.4open.science/r/ECA_Response/Visualization_Analysis.md)  for details. In this analysis, we conducted a statistic of correlation of the **average agent credits** and the **gaps between average team health points and enemy health points**. We found that 63.29% of the collected episodes suggest a positive correlation with a p-value less than 0.01. This result shows that our agent credits are closely related to team and enemy health points. The reason is that the agent credits in ECA are computed based on the episodic returns, and the team needs to eliminate all the enemies and earn the final reward (the only reward equal to the episodic return). For the episodes that suggest a negative correlation, we find that their p-value is usually greater than 0.01, which means that the negative correlation is insignificant.
>
> As the paper addresses explicit credit assignment, related works introduced in Section 5.1 should be compared more thoroughly.
>
> - We compared ECA to COMA on 5m_vs_6m, 3s5z_vs_3s6z, and MMM2. The results are as follows:
>
> [COMA-5m_vs_6m](https://anonymous.4open.science/r/ECA_Response/COMA-5m_vs_6m.png)
> [COMA-3s5z_vs_3s6z](https://anonymous.4open.science/r/ECA_Response/COMA-3s5z_vs_3s6z.png)
> [COMA-MMM2](https://anonymous.4open.science/r/ECA_Response/COMA-MMM2.png)
>
> - We select these tasks since the policy needs to control more agents, and the task is more difficult than relative tasks, such as 3s_vs_5z. The implementation of COMA comes from an [open-source repository](https://github.com/jk96491/SMAC). Within 2M environmental steps and three times repeat on each task, we find that the policies learned by COMA achieve similar winning rates as ECA on 3s5z_vs_3s6z. The trends show that COMA is not converged, and it is promising to achieve higher performance. However, COMA struggles on 5m_vs_6m and MMM2, where the winning rates of learned policies are around 0. Therefore, we infer that ECA is more stable than COMA on the selected tasks.
>
> - For other related works, we do not select them as baselines since they were thoroughly compared by previous works or lack of public implementation. Moreover, we continue experimenting with COMA on more tasks and are open to adding baselines for comparison if reviewers require it.
>
> ## We appreciate the reviewers' efforts and advice. We have added the requested changes to the paper and the replies.

---

### Decision · Action_Editor_NLRw · 2025-05-14

**Recommendation:** Reject

**Comment:**

Several drawbacks exist in the manuscript.

1. The proposed method may hinder exploration, which is a critical aspect in MARL scenarios.
2. The method heavily depends on the clustering of states. However, there may be cases where similar states play very different and important roles. In such scenarios, the proposed method may fail to perform effectively.
3. The experimental results are not as strong or significant as those reported in the most recent works.
4. Sloppy presentation and writing
5. Missing baselines. The paper lacks thorough comparisons with some important and widely-used baselines, such as COMA and SQDDPG, which focus on explicit credit assignment. The authors have mentioned that they are unable to compare to other works due to a lack of public implementations, however several algorithms cited in Section 5.1 are standard MARL baselines with implementations readily available in open-source packages like pymarl and JaxMARL.

Therefore, I conclude rejection for the submission.

**Audience:**

Yes, some individuals in TMLR's audience will be interested in knowing the findings of this paper.

**Claims And Evidence:**

The experiments do not showcase the advertised contributions of the paper. Therefore, the results provided do not indicate that the proposed method provides better credit assignment than existing methods.